# Preclinical Development of FA5, a Novel AMP-Activated Protein Kinase (AMPK) Activator as an Innovative Drug for the Management of Bowel Inflammation

**DOI:** 10.3390/ijms22126325

**Published:** 2021-06-13

**Authors:** Luca Antonioli, Carolina Pellegrini, Matteo Fornai, Laura Benvenuti, Vanessa D’Antongiovanni, Rocchina Colucci, Lorenzo Bertani, Clelia Di Salvo, Giorgia Semeghini, Concettina La Motta, Laura Giusti, Lorenzo Zallocco, Maurizio Ronci, Luca Quattrini, Francesco Angelucci, Vito Coviello, Won-Keun Oh, Quy Thi Kim Ha, Zoltan H. Németh, Gyorgy Haskó, Corrado Blandizzi

**Affiliations:** 1Department of Clinical and Experimental Medicine, University of Pisa, 56126 Pisa, Italy; mfornai74@gmail.com (M.F.); laura.benvenuti962@gmail.com (L.B.); v.dantongiovanni@gmail.com (V.D.); clelia.disalvo.pi@gmail.com (C.D.S.); g.semeghini@studenti.unipi.it (G.S.); c.blandizzi@gmail.com (C.B.); 2Department of Pharmacy, University of Pisa, 56126 Pisa, Italy; carolina.pellegrini87@gmail.com (C.P.); concettina.lamotta@unipi.it (C.L.M.); l.zallocco@gmail.com (L.Z.); luca.quattrini@farm.unipi.it (L.Q.); Francesco.angelucci@uib.no (F.A.); vito.coviello@for.unipi.it (V.C.); 3Department of Pharmaceutical and Pharmacological Sciences, University of Padova, 35122 Padova, Italy; rocchina.colucci@unipd.it; 4Department of Translational Research and New Technologies in Medicine and Surgery, University of Pisa, 56126 Pisa, Italy; lorenzobertani@gmail.com; 5School of Pharmacy, University of Camerino, 62032 Camerino, Italy; laura.giusti@unicam.it; 6Department of Pharmacy, University “G. d’Annunzio” of Chieti-Pescara, 66100 Chieti, Italy; maurizio.ronci@unich.it; 7Research Institute of Pharmaceutical Sciences, College of Pharmacy, Seoul National University, Seoul 151-742, Korea; wkoh1@snu.ac.kr (W.-K.O.); htkquy@ctu.edu.vn (Q.T.K.H.); 8Department of Anesthesiology, Columbia University, New York City, NY 10027, USA; zhnemeth@gmail.com (Z.H.N.); gh2503@cumc.columbia.edu (G.H.); 9Department of Surgery, Morristown Medical Center, Morristown, NJ 07960, USA

**Keywords:** inflammatory bowel diseases, immune system, AMPK, DNBS colitis, oxidative stress

## Abstract

Acadesine (ACA), a pharmacological activator of AMP-activated protein kinase (AMPK), showed a promising beneficial effect in a mouse model of colitis, indicating this drug as an alternative tool to manage IBDs. However, ACA displays some pharmacodynamic limitations precluding its therapeutical applications. Our study was aimed at evaluating the in vitro and in vivo effects of FA-5 (a novel direct AMPK activator synthesized in our laboratories) in an experimental model of colitis in rats. A set of experiments evaluated the ability of FA5 to activate AMPK and to compare the efficacy of FA5 with ACA in an experimental model of colitis. The effects of FA-5, ACA, or dexamethasone were tested in rats with 2,4-dinitrobenzenesulfonic acid (DNBS)-induced colitis to assess systemic and tissue inflammatory parameters. In in vitro experiments, FA5 induced phosphorylation, and thus the activation, of AMPK, contextually to the activation of SIRT-1. In vivo, FA5 counteracted the increase in spleen weight, improved the colon length, ameliorated macroscopic damage score, and reduced TNF and MDA tissue levels in DNBS-treated rats. Of note, FA-5 displayed an increased anti-inflammatory efficacy as compared with ACA. The novel AMPK activator FA-5 displays an improved anti-inflammatory efficacy representing a promising pharmacological tool against bowel inflammation.

## 1. Introduction

Crohn’s disease and ulcerative colitis, the two major forms of inflammatory bowel disease (IBD), are severe chronic pathologies characterized by a wide range of gastrointestinal and extra-digestive symptoms with a substantial negative impact on patient well-being [1]. Although the exact cause of IBDs remains still undetermined, available evidence indicates a deregulated immune response towards the commensal bacterial flora as responsible for intestinal inflammation in genetically predisposed individuals [1].

In particular, the enteric inflammatory response observed in IBD patients appears to be driven by an altered metabolic activity of gut epithelial lining, leading to severe intestinal barrier dysfunctions [2]. A large body of evidence highlighted that such alteration in tissue metabolism results from a combination of marked inflammatory cell recruitment (neutrophils and monocytes) in parallel with a high proliferation rate among lymphocyte populations [3]. This increased recruitment of immune cells coincides with a massive release of Th_1_/Th_17_- or Th_2_/Th_9_-related pro-inflammatory cytokines in Crohn’s disease or ulcerative colitis, respectively [4,5]. Increased reactive oxygen species (ROS) production and a defective pro-resolutive cytokine generation (i.e., transforming growth factor (TGF)-*β* and IL-10) are also prominent features [6].

Recently, more in-depth investigations have allowed us to better understand the molecular mechanisms underlying the pathogenic mechanisms involved in IBDs, and have paved the way toward novel therapies, mainly focused to curb the activity of key inflammatory cytokines (i.e., anti-TNF or anti-IL12/IL-23) or to dam immune cell homing (vedolizumab) [7]. Although these pharmacological approaches have made it possible to control the progression of IBDs, in a portion of the patient population they do not ensure adequate improvement/remission, even inducing in some cases severe adverse reactions [7]. However, the perspective for innovative IBD therapies is changing. In recent years, novel pharmacological approaches to manage chronic inflammatory disorders, including IBDs, have shifted focus toward the modulation of the immune cell metabolism [8].

In this regard, several authors reported a key role of the AMP-activated protein kinase (AMPK), a heterotrimeric kinase complex, in regulating immune cell “metabolic plasticity” [9]. This drives the polarization of immune cell metabolism from a glycolytic and thus pro-inflammatory activity, toward an oxidative immunoregulatory phenotype [9]. Indeed, a reduced AMPK expression/activity has been observed in several immuno-inflammatory disorders, including IBDs [10,11]. When considering intestinal inflammation, Bai et al. (2010) reported an involvement of AMPK in the pathophysiology of experimental colitis, showing a beneficial effect of the commercially available AMPK activator acadesine in DSS-or TNBS-treated mice [12,13], leading them to hypothesize the pharmacological AMPK activation as a viable way to manage IBD patients.

Despite such promising beneficial effects of acadesine, it is worth noting that this drug displays some pharmacodynamic limitations, precluding its potential therapeutic development. Indeed, acadesine, beyond the fact that it is prone to quick catabolism by xanthine oxidase [10], also requires bioactivation by adenosine kinase [14], an enzyme that is downregulated at inflammatory sites [15], with a reduction of drug bioavailability.

Based on this background, the present study was aimed at designing and synthesizing a novel AMPK activator endowed with improved pharmacodynamic properties in comparison with acadesine, and efficacy in curbing intestinal inflammation in a rat model of DNBS colitis.

## 2. Results

### 2.1. Western Blot Analysis

In the first set of experiments, the cytotoxicity of novel compounds LA12, LA14, FA5, FA6, BP19, and BP22 was determined to estimate the optimum concentrations not toxic to C2C12 cells. The stimulatory effects on phosphorylated AMPK (Thr172) of LA14, FA5, FA6, BP19, BP22 (10 μM), and 2 μM of LA12 as concentrations without cytotoxicity (data not shown), respectively, were evaluated in differentiated mouse C2C12 skeletal myoblasts model (Appendix A). The incubation of C2C12 cells with LA14, FA5, BP19, and BP22 significantly increased the phosphorylation of AMPK at a concentration of 10 μM, whereas the total AMPK expression was not changed (Appendix A). In addition, stimulated phosphorylation of AMPK induced by FA5, BP19, and BP22 were compared with berberine as a standard AMPK activator (Appendix A).

Some studies reported that stimulation of phosphorylated AMPK (Thr172) is potentially related to the increase in SIRT 1 activity [16]. Therefore, using a Sirt1-p53 luciferase cell-based assay we evaluated the effect of novel compounds on SIRT 1 activation, showing that only FA5 significantly increased SIRT 1 activity (Appendix A). Based on these data, we focused our attention on FA5, as a candidate with the potential for stimulation of phosphorylated AMPK (Thr172) associated with SIRT1 activation.

### 2.2. Protein Phosphorylation in Protein Extracts from C2C12 Cells

Based on Western blot (WB) analysis displaying an increased AMPK phosphorylation in C2C12 cell lines, we performed a proteomic analysis to investigate in-depth the change of phosphoproteome induced by FA5. Protein extracts were obtained from three independent experiments of control and FA5-treated cells performed in duplicate. The protein phosphorylation level was first assayed by 1DE separation followed by WB analysis using a specific phospho-threonine antibody, and a global change of phosho-immunoreactive bands was observed after FA5 treatment (data not shown). Then, identification of specific phosphorylated proteins was performed and aliquots of cellular protein extracts were analyzed by coupling two-dimensional electrophoresis (2DE) and WB (Figure 1A–C). Staining of immunoblot membranes with Ruthenium II tris (bathophenanthroline disulfonate) tetrasodium salt (RuBP) before the addition of the specific anti-serine phosphorylated peptides allow us to obtain two images, the first containing all transferred proteins and the second with immunoreactive spots, which can be exactly matched. Each spot intensity was normalized by the total protein intensity as visualized in the image of the membrane stained with RuBP.

Overall, we detected 76 spots differentially phosphorylated with fold change > 2, of which 54 were over phosphorylated after treatment with FA5. All differentially phosphorylated spots were identified by LC-MS/MS and collapsed in 55 proteins (of which 34 over phosphorylated and 21 under phosphorylated). The list of proteins with their accession number, molecular weight (MW), isoelectric point (pI), and mass parameters (coverage, number of unique peptides) are shown in Table 1. For some spots (spot number: 1408, 1409, 1060a, 494, 302, 917, 1313, 526, 315, 296, 1335), more than one identification was suggested as undistinguishable according to their MW and pI.

### 2.3. Gene Ontology and Pathways Analysis

To determine the biological relevance of differentially phosphorylated proteins, we carried out a gene ontology analysis using Panther. Concerning the biological processes, most proteins were assigned to the cellular process (49.1%) and metabolic processes (32.7%) clusters. The protein class analysis revealed that most proteins were nucleic acid-binding proteins (20%) and protein modifying enzymes (17%). Minor groups of proteins also clustered in metabolite interconversion enzyme (14.3%), cytoskeletal proteins (14.3%), translational protein (11.4%), and scaffold/adaptor protein (8.6%).

Then, the quantified phosphoproteome data were included in the Ingenuity Pathways Analysis (IPA) to evaluate the signaling networks regulated through FA5 treatment. The software generates two main networks, the “cancer, gastrointestinal disease, hepatic disease” and the “post-translational-modifications, protein folding, cancer” with 57 and 50 score values, respectively. Moreover, to search for the signaling molecules that could function upstream of the regulated phosphoproteins, we performed a prediction analysis of upstream kinases/regulators and of their activity state by correlating literature reported effects with observed phosphoprotein expressions. Four kinases (protein DBF4 homolog A (DBF4), casein kinase II subunit alpha (CSNK2A1), cAMP-dependent protein kinase catalytic subunit alpha (PRKACA), and cell division cycle 7-related protein kinase (CDC7)) appeared to be the most significant upstream regulators with *p*-values of 0.00085, 0.0023, 0.0034, and 0.0037, respectively. In addition to the kinases named above, also fasudil, an isoquinoline sulfonamide drug, was observed to be a significant upstream regulator. In this regard, Yoshikawa et al. suggested fasudil to counteract inflammatory cytokine effects through ROCK inhibition and to prompt its role in preventing colon carcinogenesis in patients with inflammatory bowel diseases [17].

Finally, to generate possible causal networks which explain observed phosphorylation expression changes, hidden connections in upstream regulators were also uncovered and the presence of AMPK among participant regulators was observed. These casual networks were shown to be master regulators of transient receptor potential vanilloid 1 (TRPV1), phosphatidylinositol 3,4,5-trisphosphate 3-phosphatase (TPTE2), and polyphosphoinositide phosphatase (FIG4). Controversial results have been reported concerning the anti-inflammatory effects induced by antagonists and agonists of TRPV1 in inflammatory bowel diseases, despite their clinical efficacy are currently investigated [18,19,20]. Noteworthy, FIG4 has a role as a regulatory complex of both the synthesis and turnover of phosphatidylinositol 3,5-bisphosphate (PtdIns(3,5)P2), whose metabolism has been suggested to be an unexpected and critical link between membrane trafficking in intestinal epithelial cells and the pathogenesis of inflammatory bowel disease [21].

### 2.4. In Vivo Experiments

#### 2.4.1. Body and Spleen Weight

Measurement of body and spleen weight was assumed as indexes of systemic inflammation [22]. Six days after DNBS administration, rats displayed a significant decrease in their bodyweight in comparison with control animals (Figure 2A). Treatment with FA5 (0.5–30 mg/kg/day) or acadesine (10 mg/kg/day) induced a significant increase in the bodyweight of rats with colitis, whereas animals subjected to dexamethasone (0.1 mg/kg/day) administration did not have increased weights (Figure 2A). The induction of colitis was also characterized by a significant increase in spleen weight. This increment was dose-dependently reduced by FA5 or by acadesine, as well as by the dexamethasone administration (Figure 2B).

#### 2.4.2. Colonic Length and Macroscopic Damage

Six days after DNBS administration, inflamed rats were characterized by a shortening of colonic length, as compared with control animals. Treatment of inflamed rats with FA5 (0.5–30 mg/kg/day), acadesine (10 mg/kg/day), or dexamethasone (0.1 mg/kg/day) attenuated significantly the decrease in colonic length (Figure 3A).

The administration of DNBS was characterized by colonic thickening and ulcerations, with marked areas of transmural inflammation. In addition, adhesions and bowel dilations were detected, with macroscopic damage accounting for 7.8 ± 0.6. In this setting, rats treated with FA5, acadesine, or dexamethasone showed a significant reduction in the macroscopic damage score (Figure 3B).

#### 2.4.3. Histological Analysis

Histological examination of colonic specimens from DNBS-treated rats showed an evident inflammatory process associated with large areas of mucosal necrosis, where the glandular architecture was completely unsettled. Submucosa was thickened owing to the presence of edema and marked infiltration with inflammatory cells associated with vasodilation. The mucosa and submucosa surrounding the necrotic area exhibited inflammation associated with marked cellular infiltration, compared with tissues obtained from control animals. Colonic sections from inflamed rats displayed severe damage 6 days after DNBS treatment (Figure 4). Acadesine (10 mg/kg/day) and FA5 reduced the histological damage in colonic tissues obtained from inflamed rats (Figure 4). Dexamethasone (1 mg/kg) determines a significant improvement in microscopic scores (Figure 4).

#### 2.4.4. Western Blot Analysis

*Evaluation of AMPK activation*. Western blot analysis showed a detectable expression of AMPK in colonic tissues from control rats (Figure 5). In tissues from animals with colitis, the expression of AMPK was significantly reduced (Figure 5). Such a decrease was counteracted by the administration of the novel AMPK activator FA5 (Figure 5).

*Effect of FA5 on tight junction expression*. Colonic mucosal scrapings from control animals and DNBS treated rats were assayed for zonulin-1 (ZO-1), occludin, and claudin-1. Control rats displayed the presence of both ZO-1, occludin, and claudin-1 in their colonic mucosa (Figure 6). The induction of experimental colitis induced a significant decrease in the tight junctions compared to control animals (Figure 6). The pharmacological activation of AMPK with FA5 determined a normalization of occludin expression in colonic tissues from rats with colitis (Figure 6). By contrast, no significant effect of FA5 was observed on ZO-1 or claudin (Figure 6)

#### 2.4.5. Tissue TNF and IL-10 Levels

Colonic injury induced by DNBS was associated with a significant increase in tissue TNF levels in comparison with control rats (from 5.2 ± 1.0 to 9.4 ± 1.3 pg/mg). Treatments with FA5 (0.5–30 mg/kg/day), acadesine (10 mg/kg/day), or dexamethasone (0.1 mg/kg/day) significantly decreased the concentration of this pro-inflammatory cytokine (Figure 7A). Moreover, the induction of colitis was also followed by a marked decrease in the anti-inflammatory cytokine IL-10 (normal: 9.6 ± 1.7 pg/mL; DNBS: 4.2 ± 0.8 pg/mL). This reduction was counteracted significantly by FA5, acadesine, or dexamethasone administration (Figure 7B).

#### 2.4.6. MDA Assay

MDA levels in colonic tissue from control rats were 160 ± 32.5 μmol/mg tissue and resulted markedly increased in animals with colitis (582 ± 38.5 μmol/mg tissue). Treatment with all test drugs attenuated significantly the increase in tissue MDA levels elicited by colitis induction (Figure 7C).

## 3. Discussion

In recent years, the perspective for designing innovative IBD therapies is changing, readdressing the attention from a paradigm aimed at “fighting inflammation,” mainly via non-biological (i.e., aminosalicylates, thiopurines, and steroids) or biological therapies (mainly acting on TNF, IL-23, or adhesion molecules), toward a novel pharmacological approach aimed at “targeting and advancing inflammation resolution” [23]. Indeed, there is increasing evidence to suggest that a failure in the “mediators of resolution” of inflammation may underpin autoimmune and inflammatory diseases, including IBDs [23]. Based on these premises, increasing efforts have been addressed toward the characterization of such pro-resolving mechanisms allowing to identify novel molecular targets useful to design resolution-based therapies for IBDs [23].

Among them, the AMP-activated protein kinase (AMPK) represents a critical molecular target involved in reprogramming the immune cell populations from a pro-inflammatory glycolytic metabolism towards a more oxidative phosphorylation which characterize the pro-resolutive phenotype of immune cells [24]. The genetic or the pharmacological in vitro ablation of AMPK elicited a proinflammatory polarization of antigen-presenting cells [25], an increased activity of cytotoxic T lymphocytes and NK cells [26], and an imbalance between T_H1_/T_reg_ cells [27], indicating the AMPK as a “master switch” of immune cell polarization to an anti-inflammatory functional phenotype [10].

At present, a set of preclinical experiments revealed a beneficial effect of acadesine, a commercially available AMPK activator in a mouse model of acute and relapsing TNBS-induced colitis [12,13]. However, acadesine displays some pharmacodynamic limitations: (i) it needs to be bioactivated by the enzyme adenosine kinase, which undergoes a marked down-regulation at inflammatory sites [10]; (ii) a reduced drug bioavailability resulting from a quick degradation via xanthine oxidase, limiting the potential therapeutical applications of acadesine [10]. It is worth noting that acadesine, being a mimic of AMP, exerts also AMPK-independent off-target effects [28], thus spurring the need to develop novel and more targeted AMPK activators.

The aims of the present study were: (a) to design and synthesize a novel AMPK activator (named FA5) endowed with improved pharmacodynamic properties in comparison with acadesine; (b) to evaluate, via in vitro experiments, the ability of FA5 to activate AMPK; (c) to perform a proteomic analysis aimed at evaluating the changes of the phosphoproteome induced by the novel AMPK activator FA5; (d) to compare the efficacy of FA5 with the commercially available AMPK activator acadesine in an experimental model of bowel inflammation.

The novel AMPK activator, 3-amino-5-phenylbenzofuran-2-carboxamide (FA5; Appendix A) was designed by referring to the crucial pharmacophoric residues of acadesine [29], represented by the 4-carboxamide residue and the N3 atom of the five-membered heterocyclic ring. The 2,3-disubstituted furan fragment, chosen as a bioisostere of the imidazole scaffold, perfectly recalls the key structural elements of the reference compound, both from an electronic and a steric point of view. Moreover, the planar and aromatic phenyl core in place of the ribose portion provides high lipophilicity to the novel derivative while reducing consistently its topological polar surface area, thus ensuring in principle a favorable bioavailability profile.

In a set of in vitro experiments performed on C2C12, a murine myoblast cell line stably expressing AMPK [30,31,32], the incubation with FA5 induced phosphorylation, and thus the activation, of this enzyme. Interestingly, in the same cell line we observed that contextually to the activation of AMPK, FA5 elicited also a significant activation of SIRT-1. This evidence is in line with previous reports describing that the pharmacological stimulation of AMPK is followed by SIRT-1 activation as well as by the stimulation of mitochondrial biogenesis and function [33]. In the last years, the AMPK/SIRT-1 enzyme axis displayed a crucial role in counteracting the immune-inflammatory processes, blunting the activity of the pro-inflammatory transcription factors Stat3 and NF-κBp65, c-Jun (a component of the transcription factor AP-1), and Smad7, an inhibitor of TGF-β1 activity [34]. In this regard, a down-regulation has been described of the AMPK/SIRT-1 axis in the gut of IBD patients as well as in mice with an IBD-like colitis, also reporting the abrogation of pro-inflammatory cytokine production and a reduction of the intestinal damages following the treatment of mice with an AMPK or SIRT-1 activators [11,34].

The in vitro investigation was corroborated by a set of phosphoproteomic experiments, which confirmed the efficacy of FA5 in activating the AMPK enzyme complex as well as in stimulating SIRT-1. In the same set of experiments, we observed that the AMPK activator FA5 determined also a reduction in the downstream pathway ROCK, MAPK, and RhoA in C2C12. This is an intriguing point since the RhoA/ROCK signaling is tightly associated with the level of immune system activation and the production of pro-inflammatory factors [35]. Interestingly, Horowitz et al. reported an increased RhoA/ROCK in the bioptic samples from IBD patients when compared with healthy intestinal tissues [36]. In particular, a critical involvement of the RhoA/ROCK also in the epithelial barrier dysfunctions has been reported, affecting the apical junctional complex and thus epithelial barrier integrity, under adverse intestinal conditions (i.e., IBDs, celiac disease) [37].

In the last years, several studies have demonstrated evidence of mitochondrial stress and alterations in mitochondrial function within the immune cells and the intestinal epithelium of patients with IBD and mice undergoing experimental colitis [38]. In this regard, a growing body of evidence points to a specific regulation of various aspects of mitochondrial metabolism and homeostasis by AMPK, stimulating the lipid β-oxidation and suppressing the glycolysis [39]. In line with this view, the pharmacological activation of AMPK via FA5 incubation elicited a shift of the cellular metabolism from a glycolytic metabolism to β-oxidation of fatty acids and oxidative metabolism. It is worth noting that the mitochondrial metabolic shift is critically involved in driving immune cell phenotype and their function [40]. Indeed, in pro-inflammatory cells, such as activated monocytes or T and B cells, energy is generated by increasing glycolysis [41], while in regulatory cells, such as T_reg_ cells or M2 macrophages, energy is generated by increasing mitochondrial function and beta-oxidation [42,43]. Evidence has reported that metformin, a first line of oral antidiabetic treatment, suppresses immune responses mainly through its direct effect on the cellular functions of various immune cell types by induction of AMPK and subsequent inhibition of mTORC1, and by inhibition of mitochondrial ROS production [44,45]. Metformin, interfering with the T helper 17/regulatory T cell balance, the germinal center formation, the autoantibody production, the macrophage polarization, the cytokine synthesis, neutrophil extracellular traps release, restored immune homeostasis, and improved disease severity in several models of inflammatory diseases [44].

The third part of the present study has been designed to evaluate the putative beneficial effect of FA5 administration in a rat model of experimental colitis. To pursue this aim, the experiments were performed in a rat model of DNBS-induced colitis, resembling human Crohn’s disease in terms of macroscopical (bodyweight loss, diarrhea, ulceration, and bleeding), histological, and immunological features including depletion of goblet cells and transmural infiltration of polymorphonuclear cells and predominant NF-κB-dependent T_h1_ activation [46]. The DNBS-induced colitis represents a useful experimental model to test the putative beneficial effects of novel anti-inflammatory compounds endowed with a potential therapeutic efficacy in human IBDs [47]. The suitability of this model was previously validated by our research group, confirming the efficacy of dexamethasone in ameliorating systemic and tissue inflammatory indexes [48].

In this set of experiments, we observed that the presence of intestinal inflammation is characterized by a reduction of AMPK expression in colonic tissues. This is in line with previous reports highlighting a marked reduction of AMPK phosphorylation in murine models of colitis [11,49,50,51] as well as in patients with IBDs [11]. Of note, our novel AMPK activator FA5 induced a significant activation of the AMPK enzyme complex. Based on these premises, we evaluated the efficacy of FA5 in counteracting the bowel inflammation in DNBS-induced colitis. In particular, we observed that in rats with colitis, FA5 and acadesine administration counteracted the bodyweight loss and the increment of spleen weight, regarded as systemic inflammatory indexes. In contrast, dexamethasone showed a detrimental effect on bodyweight, in virtue of its systemic glucocorticoid activity, leading to muscular atrophy. Moreover, treatments with both AMPK activators were associated with improvements in colonic shortening and in macroscopic and histologic damage scores associated with intestinal inflammation. In parallel, our findings showed a reduction of TNF and an increase in the anti-inflammatory cytokine IL-10 in the colonic tissues from animals with colitis, treated with FA5 or acadesine.

Of note, it has been demonstrated that the disruption of the intestinal epithelial barrier represents a crucial mechanism in the onset and development of IBDs [52]. Accordingly, we observed a marked decrease in tight junction expression in rats with colitis. This is in line with preclinical and clinical evidence that showed an altered expression of the tight junction proteins, including occludin, claudins, in the presence of intestinal inflammation [53]. Of note, the pharmacological activation of AMPK with FA5 exerted a restorative effect on tight junctions in rats with DNBS colitis, showing an ameliorative effect mainly on occludin expression.

MPK activation is consistent with previous studies reporting a significant role of this enzyme in the modulation of cytokine release from immune cell populations [10]. In line with our results, Bai et al. showed that the pharmacological stimulation of AMPK with acadesine reduced NF-kB activation in colonic tissues, resulting in a marked decrease in pro-inflammatory cytokines (TNF, IL-17, IL-12, IL-23, IFN-γ) and a down-regulation of iNOS expression in intestinal macrophages [12]. Interestingly, a set of studies showed that the pharmacological stimulation of AMPK elicited the re-polarization of leukocyte phenotypes inducing programs of anti-inflammatory gene expression, including IL-10 [54,55].

In parallel, DNBS-treated rats administered with FA5 or acadesine displayed a reduction in colonic MDA levels, a lipid peroxidation marker, thus indicating an antioxidant effect of both AMPK activators. It is widely recognized that AMPK exerts a pivotal role as an oxidative stress sensor and redox regulator in addition to its traditional role as an energy sensor and regulator [56]. Indeed, once activated AMPK is essential in maintaining intracellular redox status by inhibiting oxidant production by NADPH oxidases, mitochondria, or by eliciting the expression of antioxidant enzymes (i.e., SOD-2 and UCP-2) [56,57].

In conclusion, our results expand current knowledge about the beneficial effects of FA5, a novel AMPK activator in experimental colitis. Indeed, the pharmacological modulation of AMPK represents an intriguing molecular target to develop novel therapeutic strategies aimed at supporting the resolution of intestinal inflammation, via a metabolic reprogramming of immune cells and as well as the attenuation of pro-oxidative systems. Therefore, the present observations might pave the way to the design and clinical development of novel AMPK selective activators as a useful tool to manage patients with IBDs.

## 4. Materials and Methods

### 4.1. Synthesis of 3-Amino-5-phenylbenzofuran-2-carboxamide, FA5

The test compound FA5 was synthesized at the Department of Pharmacy of the University of Pisa, Italy, following the three-step procedure depicted in Appendix A [44]. Briefly, a mixture of the commercially available 5-bromo-2-hydroxybenzonitrile (1.00 mmol), 2-bromoacetamide (1.20 mmol), and Cs_2_CO_3_ (1.20 mmol) in DMF was heated and stirred until the intermediate, 2-(4-bromo-2-cyanophenoxy) acetamide, was obtained. The intermediate was then cyclized to the corresponding 3-amino-5-bromobenzofuran-2-carboxamide by heating it in an ethanolic solution in the presence of KOH. The reaction of the bromo-derivative (1.00 mmol) with phenyl boronic acid (1.20 mmol), Pd(OAc)_2_ (0.10 mmol), and PPh_3_ (0.20 mmol) in refluxing toluene and Na_2_CO_3_ 2M yielded the target FA5. FA5 was purified by crystallization from AcOEt and characterized by physio-chemical and spectroscopic data. Yield 56%. M.p.: 163–165 °C. ^1^H-NMR (DMSOd_6_, δ, ppm): δ: 6.067 (s, 2H, exc.), 7.294 (bs, 2H, exc.), 7.376 (t, 1H, J = 7.36 Hz), 7.519–7.481 (m, 3H), 7.694 (d, 2H, J = 8.44 Hz), 7.734 (dd, 1H, J = 1.96, Hz J = 8.69 Hz).

### 4.2. In Vitro Experiments

#### 4.2.1. Cell Culture and Protein Extraction

The C2C12 cell line (mouse skeletal myoblasts) was obtained from ATCC (Manassas, VA, USA). C2C12 cell lines were grown in Dulbecco’s Modified Eagles Medium (DMEM) supplemented with 10% fetal bovine serum (FBS) and 1% penicillin/streptomycin (100 units/mL/100 µg/mL) until 80% confluence. To prepare C2C12 myotubes, the cells were incubated with DMEM supplemented with 2% horse serum (HS). To investigate the effect of the specific compound, the myotube cell monolayer was gently rinsed with phosphate buffer saline (PBS) and once with DMEM medium without serum and then incubated for 24 h (h) in a serum-free medium. In order to determine the optimal rinsing conditions and incubation time, cell viability was evaluated by the trypan blue assay dye exclusion. After starvation, myotubes were incubated with FA5 compound (10 μM) or with vehicle (DMSO) for 30 min. After the incubation time, the medium was immediately removed, the cells were washed with PBS and lysed by scraping using the rehydration solution (7 M urea, 2 M thiourea, 4% CHAPS, 60 mM DTT) added with 50 mM NaF, 2 mM Na3VO4, 10 mM glycerophosphate, and 1 μL/10^6^ cell protease inhibitors. Then, detached cells were collected, sonicated, and allowed to rehydrate for 1 h at room temperature (RT) with occasional stirring. Thereafter, the solution was centrifuged at 17,000× *g* for 15 min at RT to eliminate insoluble materials. The protein content was measured by the RC/DC assay (Bio-Rad, Hercules, CA, USA). Bovine serum albumin (BSA) was used as a standard.

#### 4.2.2. Cytotoxicity Assay

C2C12 cell lines (mouse skeletal myoblasts) were maintained in growth medium of Dulbecco’s Modified Eagle’s Medium (DMEM) supplemented with 10% fetal bovine serum (FBS) (Gibco, Waltham, Massachusetts, USA), and the cells were incubated at 37 °C and 5% CO_2_. A cytotoxicity assay was carried out using the 3-(4,5-dimethyl-2-thiazolyl)-2,5-diphenyl-2H-tetrazolium bromide (MTT) (Sigma) to measure the cell viability. Briefly, C2C12 cells were seeded onto 96-well plates at 5000 cells/well and incubated for 12 h. Then, the cells were treated with different concentrations of compounds which dissolved in serum-free medium. After 24 h of incubation, the 2 mg/mL MTT solution (20 μL) was added to each well and incubated for 4 h. Formazan was dissolved in 100 μL of DMSO and then the absorbance was measured by an Absorbance Microplate Reader (VersaMax^TM^, Randor, PA, USA) at 550 nm.

#### 4.2.3. Western Blot Analysis of AMPK Phosphorylation in Differentiated Mouse C2C12 Skeletal Myoblasts

C2C12 myoblasts were seeded onto a 6-well plate and incubated until 70–80% confluence. To prepare C2C12 myotubes, the cells were incubated with DMEM supplemented with 2% horse serum (Gibco, Waltham, MA, USA). The differentiation medium was changed every 1–2 days until the myotubes had formed. For the Western blot analysis, the myotubes were incubated with tested compounds for 30 min. After that, the cells were washed with cold PBS and lysed using the lysis buffer (50 mM Tris-HCl pH 7.6, 120 mM NaCl, 1 mM EDTA, 0.5% NP-40, 50 mM NaF, and centrifuged at 12,000 rpm for 20 min. The protein concentrations were determined using a protein assay kit (Bio-Rad Laboratories, Inc., Hercules, CA, USA). Aliquots of lysates were electrophoresed on 12% SDS-polyacrylamide gels and then the gels were electronically transferred to polyvinylidene fluoride (PVDF) membranes (PVDF 0.45 µm, Immobilon-P, Merck Kenilworth, NJ., USA). Membranes were then incubated with primary antibodies: p-AMPKα Thr^172^ (cell signaling), AMPKα (cell signaling), or mouse monoclonal actin (InvitrogenWaltham, MA, USA). After incubation with secondary antibodies, the membranes were detected using an enhanced chemiluminescence Western blot detection kit (Thermo sci., Rockford, IL, USA).

#### 4.2.4. Sirt1 Assay

*Cell Culture and Transfection.* HEK293 cells were kept in a 37 °C incubator with 5% CO_2_ and cultured in DMEM containing 10% fetal bovine serum. The cells were placed in 24-well plates with a concentration of 10^5^ cells/1 mL medium in each well. After 24 h, the cells were transfected with 0.2 μg PG13-luc plasmid, 0.2 μg RSV-β-gal plasmid, 0.1 μg myctagged p53 plasmid (myc-p53), and 0.2 μg flag-tagged SIRT1 plasmid (flag-SIRT1) using the PEI transfection reagent (Polyscience, Inc., Warrington, PA, USA). The cultures were then maintained in DMEM medium without serum and were treated with the compounds of interest after 24 h post-transfection.

*SIRT1 Deacetylation with a Luciferase Reporter Cell-Based Assay*. Cells were transfected with PG13-luc (wt p53 binding sites) and the reporter plasmid, as well as with the plasmid encoding flag−flag-SIRT1 and the plasmid encoding myc-p53, with an internal control of the RSV-β-gal plasmid. The amount of transfected DNA in each well was the same. Using an analytical luminometer (Promega, Madison, WI, USA), the luciferase activity was measured based on the addition of 30 μL of luciferin into 70 μL of lysate. A Dual-Luciferase assay kit (Promega, Madison, WI, USA) was used for checking the promoter activity based on measured renilla luciferase and luciferase. Cells were lysed and assayed for flag-SIRT1 reporter activities, namely, myc-p53 and p53, which were corrected by constitutive β-galactosidase luciferase expression. Calculation of normalized values was performed by dividing the luciferase activity by the renilla luciferase activity.

#### 4.2.5. Two-Dimensional Electrophoresis and Western Blot Analysis

2-DE coupled with Western blot (WB) was employed to detect specific phosphorylated proteins. 2-DE was performed on mouse differentiated myotubes samples untreated and treated with 10 μM FA5 compound. 2-DE was carried out essentially as previously described [20]. Briefly, 200 µg of proteins were filled up to 450 μL in rehydration solution. Immobiline Dry-Strips (GE Health Care Life Sciences, Little Chalfont, UK), 18 cm, nonlinear gradient pH 3–10 were rehydrated overnight in the sample and then transferred to the Ettan IPGphor Cup Loading Manifold (GE Health Care Life Sciences, Little Chalfont, UK) for isoelectrofocusing (IEF). The second dimension (SDS-PAGE) was carried out by transferring the proteins to 11% polyacrylamide gels.

After protein separation, 2DE immunoblotting was performed essentially as previously described [20]. For phosphoprotein detection, membranes were incubated for 2 h at room temperature with phospho-threonine primary antibody (rabbit polyclonal; 1:1000 dilution, catalog n. #9381S, Cell Signaling Technology Inc., Danvers, MA, USA). Goat anti-rabbit IgG, polyclonal antibody (HRP conjugate) (1:10,000 dilution; Enzo Life Sciences, Inc., Ann Arbor, MI, USA) was used as a secondary antibody. Immunoblots were developed using the ECL detection system (PerkinElmer, Waltham, MA, USA). The chemiluminescent images were acquired by ‘‘ImageQuant LAS4010”. For the preparative gels at the end of the second dimension, gels were stained with 1 μM Ruthenium II tris (bathophenanthroline disulfonate) tetrasodium salt (RuBP) (RuBP) (Cyanagen Srl, Bologna, Italy) in 1% phosphoric acid and 30% ethanol O/N. The software Same Spot (v4.1, TotalLab; Newcastle Upon Tyne, UK) and Image-Master 2D Platinum 6.01 (GE Health Care Life Sciences, Little Chalfont, UK) were used to match the image of the membrane stained with RuBP with the image of the membrane with phosphorylated spots and the image of the preparative gel. The protein spots of interest were cut out from the gel and identified by LC-MS/MS analysis.

#### 4.2.6. Spot Digestion and Protein Identification

Gel spots were trypsin digested and analyzed by LC-MS/MS as previously performed by [21] using a Proxeon EASY-nLCII (Thermo Fisher Scientific, Milan, Italy) chromatographic system coupled to a Maxis HD UHR-TOF (Bruker Daltonics GmbH, Bremen, Germany) mass spectrometer. Raw data were processed with DataAnalysis v4.2 to apply the lock mass calibration and then loaded into PEAKS Studio v7.5 software (Bioinformatic Solutions Inc, Waterloo, ON, Canada). The mass lists were searched against the Uniprot/Swiss-Prot database selecting Mus Musculus taxonomy (16,702 entries). Carbamidomethylation of cysteines was selected as fixed modification and oxidation of methionines, deamidation of asparagine and glutamine, phosphorylation of serine, threonine and tyrosine, and acetylation of lysines at the N-terminus were set as variable modifications. Non-specific cleavage was allowed to one end of the peptides, with a maximum of two missed cleavages and a maximum of three variable PTM per peptide. Furthermore, 10 ppm and 0.05 Da were set as the highest error mass tolerances for precursors and fragments respectively.

#### 4.2.7. Ingenuity Pathway Analysis and Panther Analysis

Gene ontology and pathway analysis were performed with QIAGEN’s Ingenuity Pathway Analysis (IPA, QIAGEN Redwood City, www.qiagen.com/ingenuity (accessed on 20 April 2020), Build version: 321501M Content version: 21249400). The analysis was performed using the core analysis option and considered only direct relationships among genes. Swiss-Prot accession number of the regulated phosphoproteins with corresponding comparison ratios and *p*-values were uploaded into the IPA software and the top canonical pathways, and upstream and master regulators associated with the uploaded phosphoproteins were listed along with the *p*-values calculated using a right-tailed Fisher’s exact test (*p* < 0.05). Additionally, gene ontology annotation was conducted by the PANTHER database (Copyright© Paul Thomas, http://pantherdb.org/, accessed on 20 April 2020).

### 4.3. In Vivo Experiments

#### 4.3.1. Animals

Albino male Sprague-Dawley rats (225–250 g bodyweight) were employed throughout the study. The animals were fed standard laboratory chow and tap water ad libitum and were not subjected to experimental procedures for at least 1 week after their delivery to the laboratory. Animal care and handling were in accordance with the provisions of the European Community Council Directive 2010/63/UE, recognized and adopted by the Italian Government. All experimental procedures were approved on 1 July 2016 by the Ethical Committee for Animal Experimentation of the University of Pisa and by the Italian Ministry of Health (authorization n° 674/2016-PR). All studies involving animals are reported in accordance with the ARRIVE guidelines for reporting experiments involving animals [58].

#### 4.3.2. Induction of Colitis and Drug Treatments

Colitis was induced in accordance with the method described previously by Antonioli et al. (2007) [22]. In brief, during a short anesthesia with isoflurane (Abbott, Rome, Italy), 15 mg of 2,4-dinitrobenzenesulfonic acid (DNBS) in 0.25 mL of 50% ethanol was administered intrarectally via a polyethylene PE-60 catheter inserted 8 cm proximal to the anus. Control rats received 0.25 mL of 50% ethanol. Animals underwent subsequent experimental procedures 6 days after DNBS administration to allow a full development of histologically evident colonic inflammation. Test drugs were administered intraperitoneally (i.p.) for 7 days, starting 1 day before the induction of colitis.

Animals were assigned to the following treatment groups: (a) FA5 (0.5, 1, 3, 10, and 30 mg/kg/day), (b) acadesine (standard AMPK activator 10 mg/kg/day), and (c) dexamethasone (0.1 mg/kg/day). DNBS-untreated animals (control group) and DNBS-treated rats (DNBS group) received the drug vehicle to serve as controls. Test drugs were dissolved in sterile DMSO, and further dilutions were made with sterile saline. The reference in vivo doses of acadesine and dexamethasone were selected on the basis of previous studies [22,59]. The macroscopic score was evaluated on the whole colon, whereas biochemical analysis was performed on specimens taken from a region of inflamed colon immediately adjacent and distal to the gross necrotic damage.

#### 4.3.3. Assessment of Colitis

At the end of the treatments, colonic tissues were excised, rinsed with saline, and scored for macroscopic and histological damage, in accordance with the criteria previously reported by Antonioli et al. (2010) [48]. The criteria for macroscopic scoring of colonic damage were as follows: (i) presence of adhesions between colonic tissue and other organs (0 none, 1 minor, 2 major adhesions); (ii) consistency of colonic fecal material (0 formed, 1 loose, 2 liquid stools); (iii) presence of ulceration (0 none, 1 hyperemia, 2 ulceration without hyperemia, 3 ulceration with inflammation at 1 side, 4 ≥ 2 sites of ulceration and inflammation, 5 major sites of damage, 6 major sites of damage extending > 2 cm). The score was then increased by 1 unit for each mm of colonic wall thickness. All parameters of macroscopic damages were recorded and scored for each rat by two observers blinded to the treatment. At the time of experiment, the colon length and the spleen weight were also measured.

#### 4.3.4. Histologic Damage Score

Histologic damage and inflammation were assessed by light microscopy on Hematoxylin/Eosin-stained histological sections obtained from whole gut specimens [48]. The histological criteria included mucosal architecture loss (0–3), cellular infiltrate (0–3), muscle thickening (0–3), crypt abscess (0, absent; 1, present), and goblet cell depletion (0, absent; 1, present). All parameters of histological damages were recorded and scored for each rat by two observers blinded to the treatment.

#### 4.3.5. Western Blot Analysis

The colon was collected from rats and flushed of fecal content with ice-cold phosphate-buffered saline (PBS), as described previously [60]. Tissues were minced and homogenized using a Potter-Elvehjem Grinder homogenizer on ice in 20% (*w/v*) TNE lysis buffer (50 mM Tris-HCl pH 7.4, 100 mM NaCl, 0.1 mM EDTA, 1% NP-40, 1% SDS, 0.1% DOC) with protease and phosphatase inhibitors. Samples were then sonicated and boiled for 5 min at 95 °C. Proteins were quantified with the Bradford assay. Total lysates were run on a 4–20% Criterion™ TGX™ Precast Midi Protein Gel (Bio-Rad, Hercules, CA, USA) and then transferred to PVDF membranes (Trans-Blot TurboTM PVDF Transfer packs, Biorad). Membranes were blocked with 3% BSA diluted in Tris-buffered saline (TBS, 20 mM Tris-HCl, PH 7.5, 150 mM NaCl) with 0.1% Tween 20. Primary antibodies against claudin-1 (sc-166338, Santa Cruz, Dallas, TX, USA), occludin (ab167161, Abcam, Cambridge, MA, USA), and ZO-1 (ab96587, Abcam) were used. Secondary antibodies were obtained from Abcam (anti-mouse ab97040 and anti-rabbit ab6721) phospho-AMPKα (BK2535T, Cell Signaling), AMPKα (BK5831T, Cell Signaling). Protein bands were detected with ECL reagents (Clarity Western ECL Blotting Substrate, Biorad). Densitometry was performed by IBright Analysis software (version, 1.2.2, Waltham, MA, USA).

#### 4.3.6. Cytokine Assays

At the time of sacrifice, colonic specimens were taken for the determination of tissue TNF (Biosource International, Camarillo, CA, USA) and IL-10 (Pierce Biotechnology, Rockford, IL, USA) levels by means of kits for enzyme-linked immunosorbent assay as previously described by Silene da Silvia et al. (2010) with minor changes. Briefly, tissue samples, previously stored at −80 °C, were weighed, thawed, and homogenized by a Polytron homogenizer (Cole Palmer Homogenizer, Vernon Hills, IL, USA) in 0.3 mL of phosphate-buffered saline, pH 7.2/100 mg of tissue at 4 °C, and centrifuged at 13,400× *g* for 20 min. One hundred-microliter aliquots of the supernatants were then used for the assay. Tissue TNF and IL-10 levels were expressed as picograms per milligram of tissue.

#### 4.3.7. Evaluation of Tissue Malondialdehyde

Malondialdehyde (MDA) concentration in colonic specimens was evaluated to obtain a quantitative estimation of membrane lipid peroxidation. Colonic tissues were weighed, minced by forceps, homogenized in 2 mL of cold buffer (20 mM Tris-HCl, pH 7.4) by a Polytron homogenizer (Cole Palmer Homogenizer), and spun by centrifugation at 1500× *g* for 10 min at 4 °C. Colonic MDA concentrations were determined by means of a kit for colorimetric assay (Calbiochem-Novabiochem Corporation, San Diego, CA, USA), and the results were expressed as micromoles of MDA per milligram of colonic tissue.

### 4.4. Drugs and Reagents

2,4-Dinitrobenzenesulfonic acid, dexamethasone, and dihydroethidium dye were purchased from Sigma Aldrich (St. Louis, MO, USA). 5-Aminoimidazole-4-carboxamide-1-beta-4-ribofuranoside (acadesine) was purchased from Cayman (Ann Arbor, MI, USA). The synthesis of FA5 was performed as reported in the present manuscript. The solutions were then frozen into aliquots of 2 mL and stored at −80 °C until use.

### 4.5. Proteomic Data Analysis

The analysis of 2-DE images was performed using the Same Spot (v4.1, TotalLab; Newcastle Upon Tyne, UK) software. The volume of each spot obtained after the detection of immunocomplexes was normalized by the total protein content obtained with RuBP staining of membranes. A comparison between cells treated with and without FA5 was performed. The significance of the differences of normalized volume for each phosphorylated spot was calculated by the paired Student’s *t*-test.

### 4.6. Statistical Analysis

The results are given as mean ± S.E.M. The significance of differences was evaluated on raw data by one-way analysis of variance followed by post hoc analysis with Student-Newman-Keuls test. *p*-values < 0.05 were considered significantly different. All statistical procedures were performed by commercial software (GraphPad Prism, version 7.0 from GraphPad Software Inc., San Diego, CA, USA).

## 5. Patents

FA5 has been patented (PCT#WO 2018189683 A1; 29 January 2019).

## Figures and Tables

**Figure 1 ijms-22-06325-f001:**
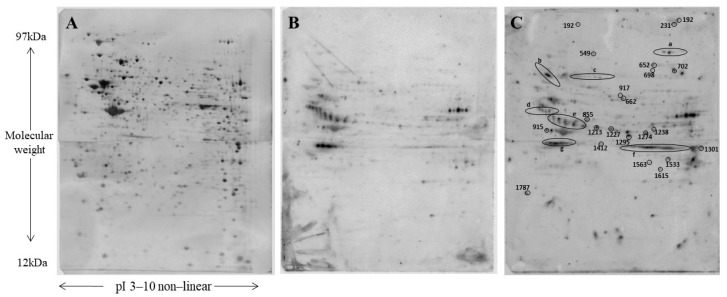
2-DE coupled with WB was employed to detect specific phosphorylated proteins. Proteins from C2C12 cells were firstly separated according to pI on Immobiline Dry-Strips (18 cm, non-linear gradient pH 3–10) and then according to molecular weight using 11% polyacrylamide gels. Subsequently, proteins were transferred onto nitrocellulose membranes. Immediately after WB, membranes were stained with RuBP. Thereafter, membranes were incubated with the anti-phosphothreonine antibody. (**A**) A representative nitrocellulose image with a 2-DE protein map of C2C12 cells (control). Proteins were detected by RuBP staining. (**B**) The same membrane with detection of the immunoreactive spots. (**C**) A representative nitrocellulose membrane with detection of the immunoreactive spots of C2C12 cells treated with FA5. Spots differentially phosphorylated are highlighted.

**Figure 2 ijms-22-06325-f002:**
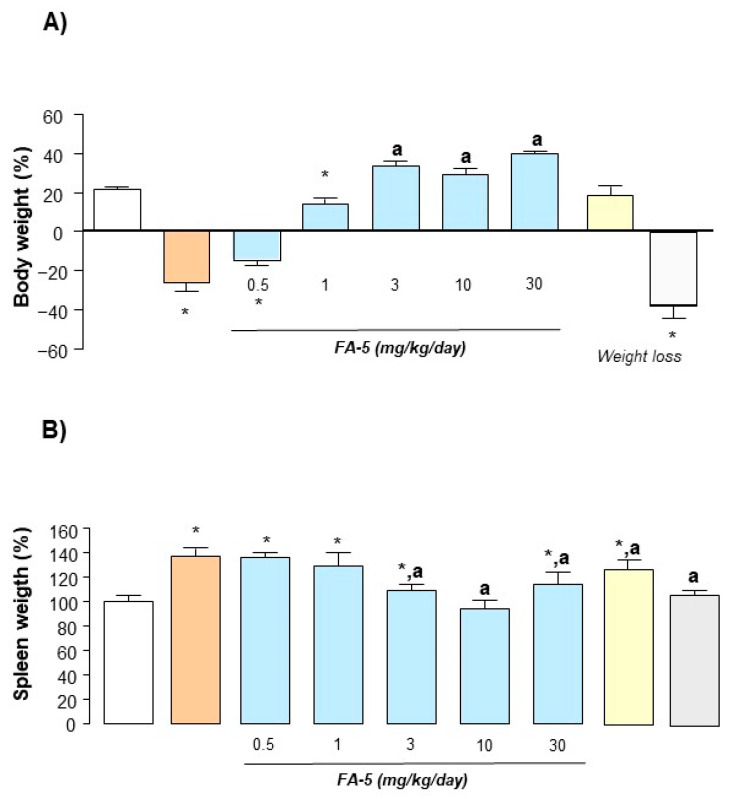
Body (**A**) and spleen weight (**B**) estimated in rats with DNBS-induced colitis treated with FA5 (0.5, 1, 3, 10, 30 mg/kg/day) acadesine (10 mg/kg/day) or dexamethasone (0.1 mg/kg/day). Each column represents the mean ± standard error of the mean (SEM) (*n* = 8). *, *p* < 0.05, significant difference versus control group; ^a^, *p* < 0.05, significant difference versus DNBS group.

**Figure 3 ijms-22-06325-f003:**
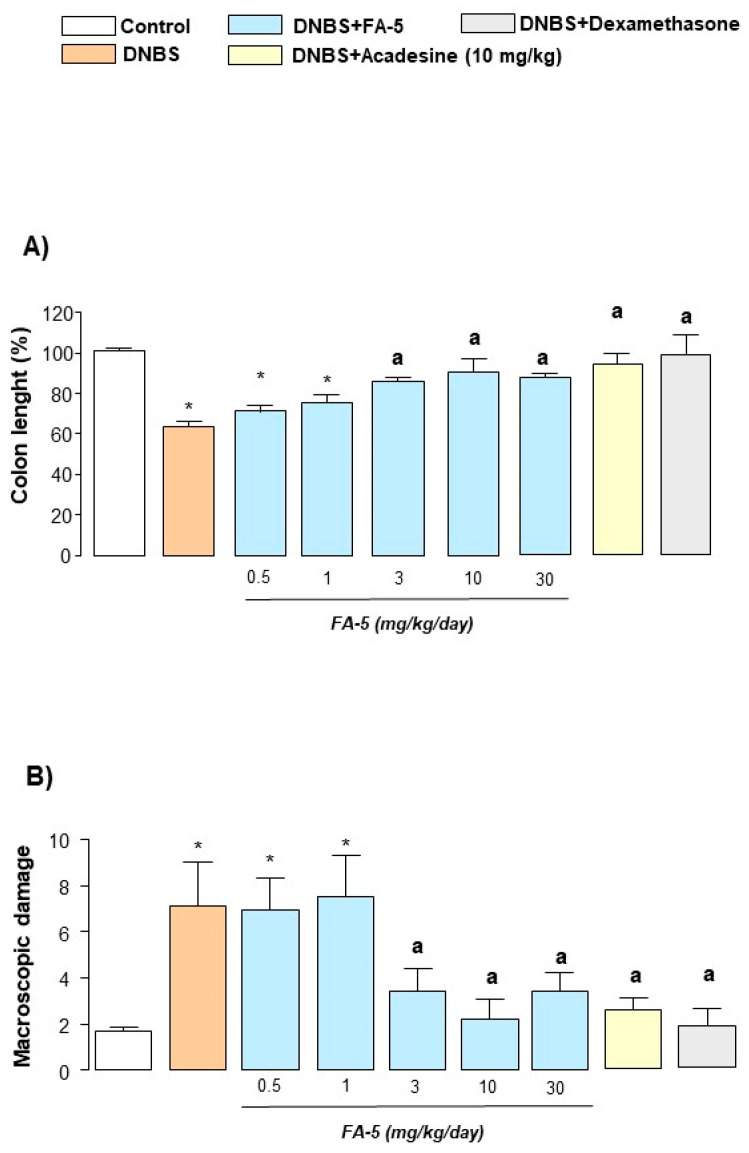
Colonic length (**A**) and macroscopic damage score (**B**) evaluated in rats with colitis, either in the absence or in the presence of FA5 (0.5, 1, 3, 10, 30 mg/kg/day), acadesine (10 mg/kg/day), or dexamethasone (0.1 mg/kg/day). Each column represents the mean ± standard error of the mean (SEM) (*n* = 8). *, *p* < 0.05, significant difference versus control group; ^a^, *p* < 0.05, significant difference versus DNBS group.

**Figure 4 ijms-22-06325-f004:**
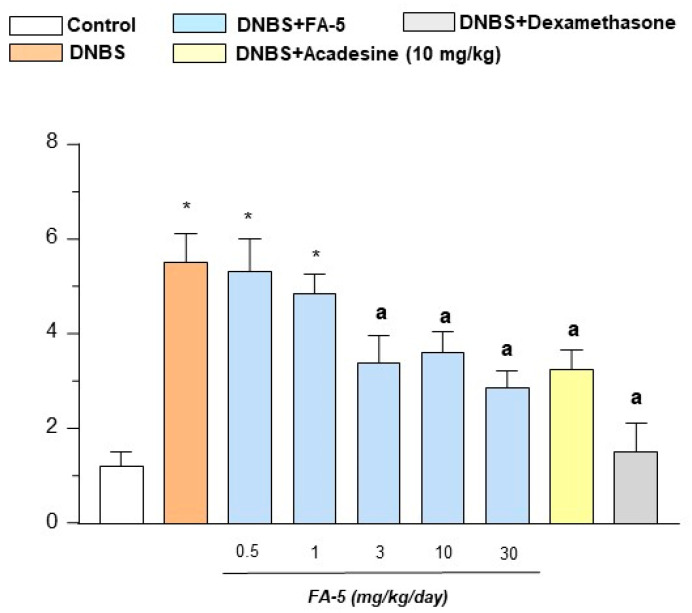
Microscopic damage score estimated for colon in rats under normal conditions or with DNBS-induced colitis, either alone or after treatment with FA5 (0.5–30 mg/kg/day), acadesine (mg/kg/day), or dexamethasone (1 mg/kg). Each column represents the mean ± standard error of the mean (SEM) (*n* = 8). * *p* < 0.05, significant difference vs. control group; ^a^ *p* < 0.05, significant difference vs. DNBS group.

**Figure 5 ijms-22-06325-f005:**
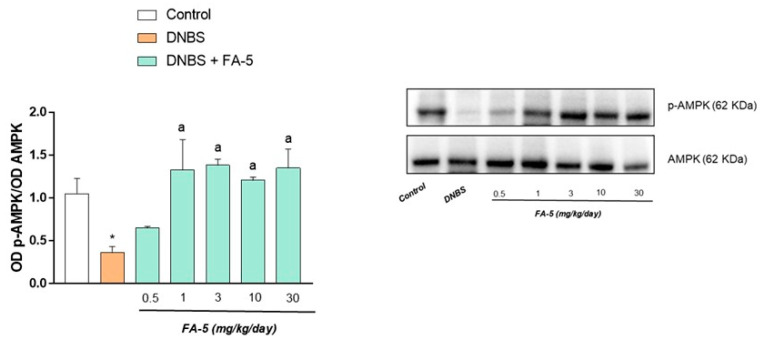
Representative blots and densitometric analysis of the expression of phospho-AMPK and AMPK in colonic tissues from control rats and animals treated with DNBS alone or in combination with FA-5 (0.5, 1, 3, 10, 30 mg/kg/day). Each column represents the mean ± standard error of the mean (SEM) (*n* = 6). One-way ANOVA followed by Tukey’s post hoc test results: * *p* < 0.05, significant difference vs. the control group; ^a^ *p* < 0.05, significant difference vs. the DNBS group.

**Figure 6 ijms-22-06325-f006:**
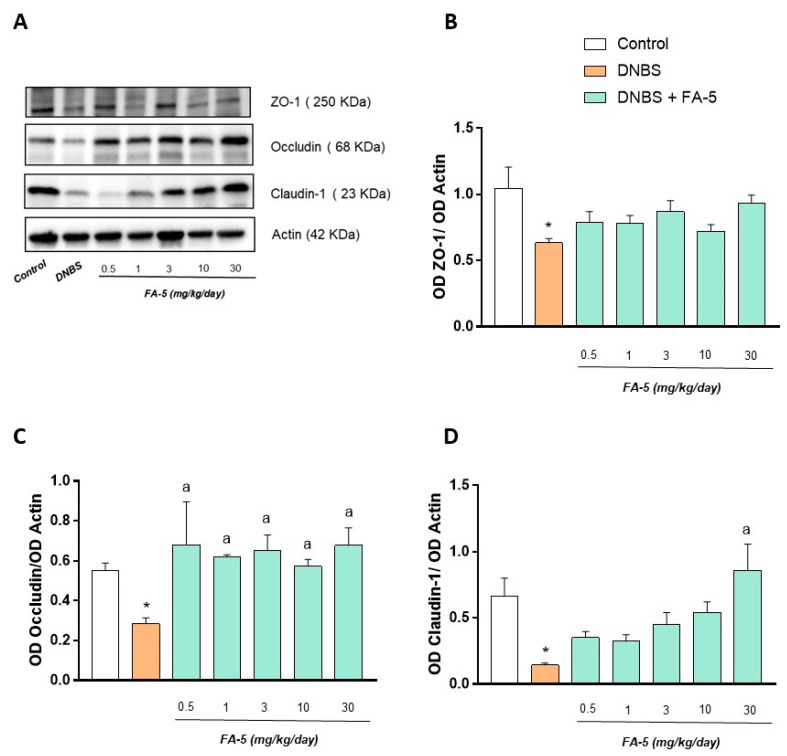
Representative blots (**A**) and densitometric analysis of the expression of ZO-1 (**B**), occludin (**C**), and claudin-1 (**D**) in colonic tissues from control rats and animals treated with DNBS alone or in combination with FA-5 (0.5, 1, 3, 10, 30 mg/kg/day). Each column represents the mean ± standard error of the mean (SEM) (*n* = 6). One-way ANOVA followed by Tukey’s post hoc test results: * *p* < 0.05, significant difference vs. the control group; ^a^ *p* < 0.05, significant difference vs. the DNBS group.

**Figure 7 ijms-22-06325-f007:**
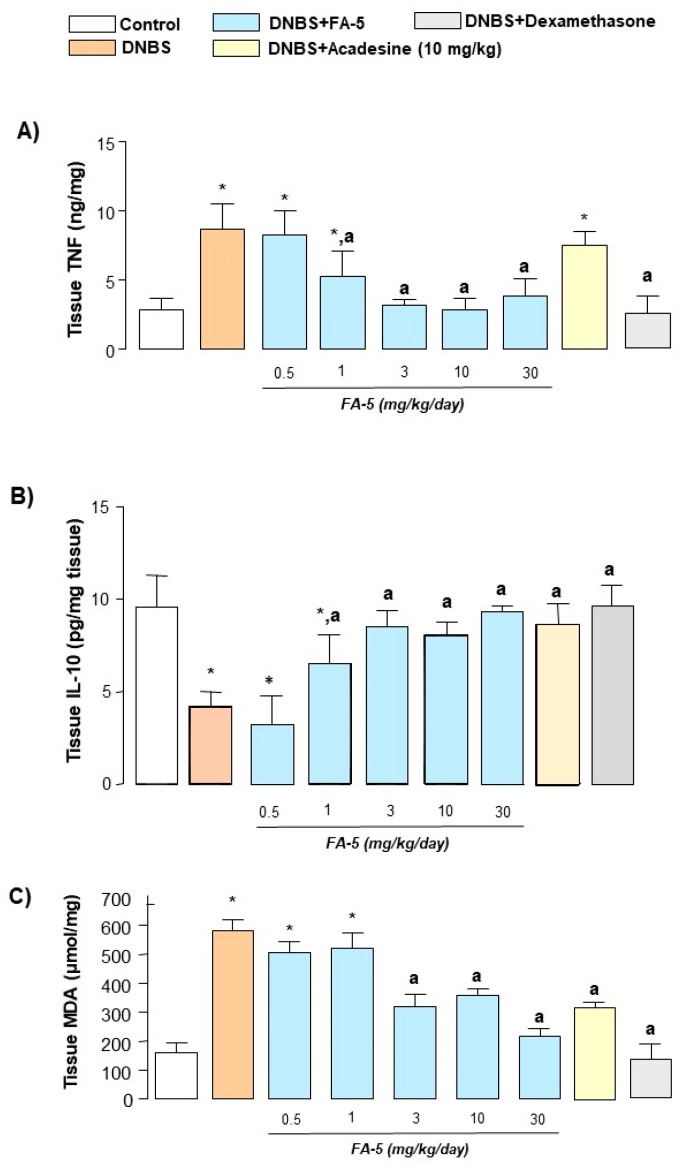
TNF (**A**), IL-10 (**B**), and MDA (**C**) levels in colonic tissues from control rats or in animals treated with DNBS either alone or in combination with FA5 (0.5, 1, 3, 10, 30 mg/kg/day), acadesine (10 mg/kg/day), or dexamethasone (0.1 mg/kg/day). Each column represents the mean ± standard error of the mean (SEM) (*n* = 8). *, *p* < 0.05, significant difference versus control group; ^a^, *p* < 0.05, significant difference versus DNBS group.

**Table 1 ijms-22-06325-t001:** List of the differentially phosphorylated protein spots identified by MS.

Spot n°/Position	ID	Protein Name	Gene Name	Coverage (%)	Unique Peptides	MW (th)	pI (th)	Ratio (FA5/CTRL)	ANOVA (*p*-Value)	Fold Phosphorylation
192	Q8CGC7	Bifunctional glutamate/proline—tRNA ligase	*Eprs*	3	4	170,078	7.75	6.3	0.03	6.3
197	Q6PDG5	SWI/SNF complex subunit SMARCC2	*Smarcc2*	2	2	132,604	5.41	4.7	0.013	4.7
201	Q02053	Ubiquitin-like modifier-activating enzyme 1	*Uba1*	21	18	117,809	5.43	2.2	0.021	2.2
231	Q8VDJ3	Vigilin	*Hdlbp*	3	4	141,742	6.43	2.5	0.023	2.5
247/a	P58252	Elongation factor 2	*Eef2*	25	21	95,314	6.41	5.2	0.002	5.2
270/a	P49717	DNA replication licensing factor MCM4	*Mcm4*	9	9	96,736	6.77	5.2	0.002	5.2
272/a	P49717	DNA replication licensing factor MCM4	*Mcm4*	5	5	96,736	6.77	5.8	0.002	5.8
279/a	P58252	Elongation factor 2	*Eef2*	25	20	95,314	6.41	3.9	0.00008	3.9
283/a	P58252	Elongation factor 2	*Eef2*	28	24	95,314	6.41	2.2	0.00009	2.2
294/a	P58252	Elongation factor 2	*Eef2*	37	35	95,314	6.41	5.8	0.002	5.8
296/a	P49717	DNA replication licensing factor MCM4	*Mcm4*	5	5	96,736	6.77	5.4	0.001	5.4
*296/a*	P58252	Elongation factor 2	*Eef2*	41	40	95,314	6.41	5.4	0.001	5.4
302	P13020	Gelsolin	*Gsn*	24	16	85,942	5.83	2.7	0.013	2.7
*302*	Q9ERG0	LIM domain and actin-binding protein 1	*Lima 1*	7	6	84,060	6.18	2.7	0.013	2.7
315	P13020	Gelsolin	*Gsn*	33	21	85,942	5.83	3.9	0.017	3.9
*315*	Q9ERG0	LIM domain and actin-binding protein 1	*Lima 1*	20	11	84,060	6.18	3.9	0.017	3.9
390	P20029	78 kDa glucose-regulated protein	*Hspa5*	58	50	72,422	5.07	4.5	0.00038	4.5
400	P14824	Annexin A6	*Anxa6*	34	20	75,885	5.34	8.2	0.003	8.2
453	P63017	Heat shock cognate 71 kDa protein	*Hspa8*	60	31	70,871	5.37	2.8	0.00006	2.8
464	Q9CZD3	Glycine—tRNA ligase	*Gars*	18	13	81,878	6.24	4.5	0.001	4.5
478	Q9CZD3	Glycine—tRNA ligase	*Gars*	26	20	81,878	6.24	6.2	0.001	6.2
492	Q9CZD3	Glycine—tRNA ligase	*Gars*	26	23	81,878	6.24	4.8	0.00005	4.8
494	P17156	Heat shock-related 70 kDa protein 2	*Hspa2*	21	2	69,642	5.5	1.63	0.00066	2.0
*494*	P63017	Heat shock cognate 71 kDa protein	*Hspa8*	59	32	70,871	5.37	1.63	0.00066	2.0
504	P48678	Prelamin-A/C	*Lmna*	15	9	74,238	6.54	0.34	0.0003	2.9
520/a	P49717	DNA replication licensing factor MCM4	*Mcm4*	16	15	96,736	6.77	6.5	0.00049	6.5
*520/a*	P58252	Elongation factor 2	*Eef2*	41	38	95,314	6.41	6.5	0.00049	6.5
521/a	P58252	Elongation factor 2	*Eef2*	26	22	95,314	6.41	4.4	0.003	4.4
526/a	P49717	DNA replication licensing factor MCM4	*Mcm4*	5	4	96,736	6.77	3.7	0.002	3.7
*526/a*	P58252	Elongation factor 2	*Eef2*	22	17	95,314	6.41	3.7	0.002	3.7
538/a	P58252	Elongation factor 2	*Eef2*	33	27	95,314	6.41	6.8	0.001	6.8
549	Q8K009	Mitochondrial 10-formyltetrahydrofolate dehydrogen	*Aldh1l2*	32	24	101,590	5.93	7.2	0.03	7.2
594	P61979	Heterogeneous nuclear ribonucleoprotein K	*Hnrnpk*	5	2	50,976	5.39	0.15	0.000086	6.5
652	Q61881	DNA replication licensing factor MCM7	*Mcm7*	33	21	81,211	5.98	2.1	0.028	2.1
662	Q91YW3	DnaJ homolog subfamily C member 3	*Dnajc3*	4	2	57,464	5.61	2.9	0.001	2.9
698	Q8R550	SH3 domain-containing kinase-binding protein 1	*Sh3kbp1*	9	5	78,170	7.15	3.5	0.018	3.5
702	P26041	Moesin	*Msn*	3	2	67,767	6.22	11	0.00057	10.9
770/d	P31324	cAMP-dependent protein kinase type II-beta regulatory	*Prkar2b*	16	5	46,167	4.9	0.41	0.00065	2.4
772/d	P99024	Tubulin beta-5 chain	*Tubb5*	30	2	49,671	4.78	0.37	0.000055	2.7
777/c	Q8C1A5	Thimet oligopeptidase	*Thop1*	36	23	78,026	5.72	2.3	0.00023	2.3
784	P56480	ATP synthase subunit beta, mitochondrial	*Atp5b*	20	8	56,301	5.19	2.1	0.00015	2.1
847	Q99L45	Eukaryotic translation initiation factor 2 subunit 2	*Eif2s2*	50	17	38,092	5.61	0.13	0.001	7.3
855	P60843	Eukaryotic initiation factor 4A-I	*Eif4a1*	34	7	46,154	5.32	2.1	0.00021	2.1
909	Q9Z1D1	Eukaryotic translation initiation factor 3 subunit G	*Eif3g*	7	2	35,638	5.7	0.47	0.00078	2.1
915	Q9JKV1	Proteasomal ubiquitin receptor ADRM1	*Adrm1*	7	3	42,060	4.94	0.45	0.001	2.4
917	P11983	T-complex protein 1 subunit alpha	*Tcp1*	51	17	60,449	5.82	2.9	0.008	2.9
*917*	P63038	60 kDa heat shock protein, mitochondrial	*Hspd1*	27	14	60,956	5.91	2.9	0.008	2.9
931	Q04447	Creatine kinase B-type	*Ckb*	34	11	42,713	5.4	0.41	0.001	2.4
1060/d	P54775	26S protease regulatory subunit 6B	*Psmc4*	45	20	47,408	5.0	0.36	0.003	2.7
1066	Q99KV1	DnaJ homolog subfamily B member 11	*Dnajb11*	28	6	40.555	5.92	0.23	0.0000009	4.3
1070/d	P20152	Vimentin	*Vim*	33	14	53,688	5.05	0.19	0.00004	5.1
1096/e	Q3UM45	Protein phosphatase 1 regulatory subunit 7	*Ppp1r7*	5	2	41,292	4.85	0.12	0.00005	8
1072/e	P20152	Vimentin	*Vim*	35	14	53,688	5.05	0.18	0.000003	5.5
1078/e	P60843	Eukaryotic initiation factor 4A-I	*Eif4a1*	23	9	46,154	5.32	0.36	0.004	2.8
1113/e	Q9Z2X1	Heterogeneous nuclear ribonucleoprotein F	*Hnrnpf*	17	6	45,730	5.3	0.36	0.003	2.7
1119/e	Q3UM45	Protein phosphatase 1 regulatory subunit 7	*Ppp1r7*	11	4	41,292	4.85	0.12	0.00005	8
1213	Q62433	Protein NDRG1	*Ndrg1*	32	8	43,009	5.69	4.9	0.00017	4.9
1227	Q62433	Protein NDRG1	*Ndrg1*	32	8	43,009	5.69	2.2	0.002	2.2
1238	P50580	Proliferation-associated protein 2G4	*Pa2g4*	33	14	43,699	6.41	2.3	0.005	2.3
1274	Q9DCL9	Multifunctional protein ADE2	*Paics*	13	6	47,006	6.94	5.0	0.043	5
1295	P35486	Pyruvate dehydrogenase E1 subunit alpha, mitoch	*Pdha1*	30	12	43,232	8.49	10	0.00005	10.4
1301	P51174	Long-chain specific acyl-CoA dehydrogenase, mitoch	*Acadl*	34	13	47,908	8.53	2.6	0.00009	2.6
1313/f	P60335	Poly(rC)-binding protein 1	*Pcbp1*	7	2	37,498	6.66	3.4	0.002	3.4
*1313/f*	Q91WK2	Eukaryotic translation initiation factor 3 subunit H	*Eif3h*	39	14	39,832	6.19	3.4	0.002	3.4
1326/f	P60335	Poly(rC)-binding protein 1	*Pcbp1*	10	3	37,498	6.66	2.7	0.001	2.7
1329/f	P60335	Poly(rC)-binding protein 1	*Pcbp1*	40	7	37,498	6.66	5.5	0.0005	5.5
1330/f	Q91WK2	Eukaryotic translation initiation factor 3 subunit H	*Eif3h*	9	2	39,832	6.19	5.7	0.00011	5.7
1335/f	Q91YR1	Twinfilin-1	*Twf1*	37	11	40,079	6.2	5.7	0.00011	5.7
1408/g	Q9CX34	Suppressor of G2 allele of SKP1 homolog	*Sugt1*	26	7	38,159	5.32	0.29	0.0001	3.4
*1408/g*	Q9R1T2	SUMO-activating enzyme subunit 1	*Sae1*	15	6	38,620	5.24	0.29	0.0001	3.4
1409/g	Q9CX34	Suppressor of G2 allele of SKP1 homolog	*Sugt1*	56	18	38,159	5.32	0.31	0.0005	3.2
*1409/g*	Q9R1T2	SUMO-activating enzyme subunit 1	*Sae1*	37	12	38,620	5.24	0.31	0.0005	3.2
1412	Q9Z0S1	3′(2′),5′-bisphosphate nucleotidase 1	*Bpnt1*	25	7	33,196	5.54	6.5	0.011	6.5
1419/g	O35295	Transcriptional activator protein Pur-beta	*Purb*	18	4	33,901	5.33	0.35	0.001	2.9
1435/f	P60335	Poly(rC)-binding protein 1	*Pcbp1*	37	9	37,498	6.66	3.0	0.001	3
1441/f	P60335	Poly(rC)-binding protein 1	*Pcbp1*	36	6	37,498	6.66	8.3	0.001	8.3
1448/f	P60335	Poly(rC)-binding protein 1	*Pcbp1*	18	5	37,498	6.66	4.7	0.013	4.7
1449/f	Q91YR1	Twinfilin-1	*Twf1*	34	11	40,079	6.2	8.6	0.003	8.6
1450/f	Q91WK2	Eukaryotic translation initiation factor 3 subunit H	*Eif3h*	37	12	39,832	6.19	5.7	0.0005	5.7
1462	P61982	14-3-3 protein gamma	*Ywhag*	43	7	28,303	4.8	0.23	0.04	4.3
1471	P63101	14-3-3 protein zeta/delta	*Ywhaz*	73	15	27,771	4.73	4.5	0.00039	4.5
1533	Q9Z130	Heterogeneous nuclear ribonucleoprotein D-like	*Hnrnpdl*	20	4	33,559	6.85	4.4	0.003	4.4
1563	Q9D883	Splicing factor U2AF 35 kDa subunit	*U2af1*	14	3	27,815	9.09	2.9	0.000535	2.9
1615	P47962	60S ribosomal protein L5	*Rpl5*	9	2	34,401	9.78	9.7	0.00034	9.7
1787	P62259	14-3-3 protein epsilon	*Ywhae*	75	23	29,174	4.63	0.32	0.00009	3.1
2489/b	P20029	78 kDa glucose-regulated protein	*Hspa5*	42	27	72,422	5.07	9.3	0.001	9.3

ID: Uniprot accession number; MW (th): theoretical Molecular Weight; pI (th): theoretical Isoelectric Point; a, b, c, d, e, f name of the area highlighted in Figure 1C where the spots are.

## Data Availability

The data presented in this study are available in this article or Appendix A.

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
