# Peer review of "Preclinical Development of FA5, a Novel AMP-Activated Protein Kinase (AMPK) Activator as an Innovative Drug for the Management of Bowel Inflammation"

_ijms, 2021, doi:10.3390/ijms22126325_

Round 1
Reviewer 1 Report
The Authors describe a novel molecule, FA-5, to manage inflammatory bowel disease (IBD). The manuscript is well written and results are highly significant and clearly reported. For these reasons, I suggest to accept it in this curreny form.
Author Response
I would like to thank the reviewer for appreciating our manuscript
Reviewer 2 Report
The manuscript by Antonioli et al evaluate the effects of FA-5 (a novel direct AMPK activator) in an experimental model of colitis in rats. The authors observe that FA-5 improved the colon length, ameliorated macroscopic damage score, and reduced TNF and MDA tissue levels in DNBS-treated rats. The study is comprehensive and generally supports the authors' conclusions. These findings may benefit from some additional clarification, as detailed below.
Comments
- Why did the authors perform the in vitro experiments using only C2C12 cell line?
- Did the authors evaluate the rectal bleeding and intestinal permeability in DNBS-treated rats?
- To evaluate the inflammatory infiltrate intestinal histology should be added.
- Please evaluate AMPK activation in DNBS-treated rats.
- To prove the anti-inflammatory phenotype the authors should add gene expression of immune cells.
- The manuscript should be edited to correct contextual and layout errors.
Author Response
- Why did the authors perform the in vitro experiments using only C2C12 cell line?
In order to proceed with a quick screening of the phosphorylative properties (and therefore activating ability) toward AMPK of our novel synthesized substances, we needed a cell line that stably expressed this enzyme complex. For these reasons, based on the available data (Egawa et al. Am J Physiol Endocrinol Metab. 2014 Feb;306(3):E344-54.; Nguyen et al. Bioorg Med Chem Lett. 2010 Jul 15;20(14):4128-31; Yan Y et al Nat Commun. 2015 Jan 30;6:6137. doi: 10.1038/ncomms7137) we have decided to use the C2C12 cell line, a murine myoblast cell line, since it stably expresses AMPK.
- Did the authors evaluate the rectal bleeding and intestinal permeability in DNBS-treated rats?
For what it concerns the evaluation of rectal bleeding, unfortunately we have not evaluated this parameter in this study.
For what it concerns the intestinal permeability we evaluated the effect of FA5 on the expression of occludin, zonulin-1 and claudin 1. The results have been introduced into the revised version of the manuscript (see lines 212-222 and lines 334-339)
- To evaluate the inflammatory infiltrate intestinal histology should be added.
As kindly requested by the Reviewer the histological evaluation of microscopic damage score has been added into the revised version of the manuscript (see lines 187-200)
- Please evaluate AMPK activation in DNBS-treated rats.
As kindly suggested by the Reviewer, we performed the evaluation of AMPK activation in control and DNBS animals as well as in rats with colitis treated with the novel AMPK activator FA5. The results have been introduced into the revised version of the manuscript (see lines 203-210)
- To prove the anti-inflammatory phenotype the authors should add gene expression of immune cells.
We fully agree with the Reviewer that the investigation of gene expression of immune cells would provide additional information about the anti-inflammatory phenotype of the cells. In this regard, we have planned to perform this kind of investigation concomitantly with a set of in vitro experiments to better characterize the ability of FA5 to reorganize toward an anti-inflammatory phenotype the immune cells. The obtained results will be object for a further and more addressed publication.
- The manuscript should be edited to correct contextual and layout errors.
As kindly indicated by the Reviewer, the revised version of the manuscript has been carefully edited to correct contextual and layout errors.
Round 2
Reviewer 2 Report
no